# Extended Reality (XR) for Condition Assessment of Civil Engineering Structures: A Literature Review

**DOI:** 10.3390/s22239560

**Published:** 2022-12-06

**Authors:** Fikret Necati Catbas, Furkan Luleci, Mahta Zakaria, Ulas Bagci, Joseph J. LaViola, Carolina Cruz-Neira, Dirk Reiners

**Affiliations:** 1Civil Infrastructure Technologies for Safety and Resilience (CITRS), Department of Civil, Environmental, and Construction Engineering, University of Central Florida, Orlando, FL 32816, USA; 2Radiology and Biomedical Engineering Department, Northwestern University, Chicago, IL 60611, USA; 3Interactive Computing Experiences Research Cluster, Department of Computer Science, University of Central Florida, Orlando, FL 32816, USA; 4Department of Computer Science, University of Central Florida, Orlando, FL 32816, USA

**Keywords:** augmented reality, virtual reality, mixed reality, extended reality, structural health monitoring, SHM, NDE, NDT, AEC

## Abstract

Condition assessment of civil engineering structures has been an active research area due to growing concerns over the safety of aged as well as new civil structures. Utilization of emerging immersive visualization technologies such as Virtual Reality (VR), Augmented Reality (AR), and Mixed Reality (MR) in the architectural, engineering, and construction (AEC) industry has demonstrated that these visualization tools can be paradigm-shifting. Extended Reality (XR), an umbrella term for VR, AR, and MR technologies, has found many diverse use cases in the AEC industry. Despite this exciting trend, there is no review study on the usage of XR technologies for the condition assessment of civil structures. Thus, the present paper aims to fill this gap by presenting a literature review encompassing the utilization of XR technologies for the condition assessment of civil structures. This study aims to provide essential information and guidelines for practitioners and researchers on using XR technologies to maintain the integrity and safety of civil structures.

## 1. Introduction

The interdisciplinary authors of this paper would like to present recent advances in structural health monitoring (SHM), mainly related to Virtual Reality, Augmented Reality, and Mixed Reality for condition assessment of civil engineering structures, as the development of such novel technologies has been progressing quite considerably over the last few years. The Civil Infrastructure Technologies for Resilience and Safety (CITRS) laboratory at the University of Central Florida has a history of exploring novel technologies for civil engineering applications starting from the late 1990s [1]. Since then, multiple studies have been carried out including the earliest investigations of novel technologies, such as the employment of computer vision [2,3,4,5] in SHM applications. Additionally, the prior research of CITRS members has demonstrated the integration of digital documentation of design, inspection, and monitoring data (collected from a civil structure) coupled with calibrated numerical models (“*Model Updating*”) [6,7]. Briefly, model updating is a methodology to update a numerically built model in real/near time based on the information obtained from the monitoring and inspection data (field data). Fundamentally, this technique is nowadays known as digital twin. Over the last decade at CITRS, the investigation of using novel methods for condition assessment and SHM applications has continued extensively. One of the earliest machine-learning-based studies was conducted for SHM applications [8,9]. Furthermore, various other new implementations followed, such as using Mixed Reality (MR) to assist the inspectors on the field [10,11] and using Virtual Reality (VR) to bring the field data to the office where parties can collaborate in a single VR environment [12]. Another recent collaborative work discusses the latest trends in bridge health monitoring [13]. Generative adversarial networks (GAN) have been recently explored in the civil SHM domain [14]. They were investigated to address the data scarcity problem [15,16,17] and used for the first time in undamaged-to-damaged domain translation applications where the aim is to obtain the damaged response while the civil structure is intact or vice versa [18,19]. Members of the CITRS group are motivated to present some of the recent advances in SHM and other notable studies from the literature.

Civil engineering is arguably one of the oldest and broadest disciplines. It employs various theories, methodologies, tools, and technologies to solve different problems in the field. With the advancements in sophisticated computer graphics and hardware, a combination of artificial intelligence (AI), Virtual Reality (VR), Augmented Reality (AR), and Mixed Reality (MR) has been utilized more in the last decade (Extended Reality (XR) is an umbrella term for VR, AR, and MR technologies). While the first VR prototype, named “Sensorama”, was introduced in 1956 by cinematographer Morton Heilig, the first AR headset was only created by Ivan Sutherland in 1968, named “The Sword of Damocles”. These two efforts marked the beginning of the new field of XR. In the 1990s and early 2000s, there were many research and commercial efforts to advance the technology, and AEC became one of the disciplines that could best benefit from this technology. However, it was not until about ten years ago that, thanks to the advances in real-time graphics hardware and increased computation power in mobile and personal computers, XR became much more widespread and more feasible as a tool that could be utilized in different workflows. In 2022, the use of XR has extended to various types of applications in different industries such as education, military, healthcare, and architecture–engineering–construction (AEC). The recent progress in XR was possible due to the advancements in and miniaturization of graphics and computational systems and the significant reduction in cost and complexity of this technology. In addition, the introduction of the concept of “metaverse”, where users experience virtual collaborative environments online, has also accelerated research and development in the XR industry in recent years. XR technologies have various advantageous uses in the AEC industry, such as training and education, structural design, heritage preservation, construction activities, and structural condition assessment. This article presents a literature review on the uses of XR technology in the condition assessment of civil structures.

### 1.1. Civil Structural Health Monitoring

As civil engineering structures age and deteriorate due to man-made or environmental stressors, condition assessment of civil structures is a critical area of research and practice. Traditionally, structural condition assessment at the local level is mostly implemented through conventional methods and visual inspection techniques, such as chain drag or hammer tapping to detect delamination and voids in concrete, or through visual inspections only [20]. To overcome the challenges of traditional methods, such as being labor-intensive, time-consuming, and subjective, structural health monitoring (SHM) [21,22] was introduced and has been applied in practice. Briefly, SHM is a methodology to examine the health status of civil structures, mainly for large and occasionally medium-sized civil structure systems, by collecting sensorial data with the use of sensors, e.g., accelerometers, strain gauges, potentiometers, cameras, lasers, and a non-destructive technique/evaluation (NDT/E). In other words, SHM can be considered a combination of structural damage diagnosis and prognosis based on analyzing and evaluating sensorial data collected from civil structures [23]. The goal of SHM is to track structural responses and possibly inputs to determine the condition of structures to support decision making for a number of purposes, such as design verification, damage detection, effective and efficient maintenance, and operations. This is achieved by tracking changes in a structure’s geometric or material properties based on the analyzed sensorial data for the following decision-making process. Generally, SHM methods are classified by their application levels at local and global levels [5]. While local methods (NDT/E and non-contact tools or NCT) are employed to diagnose the local defects, global methods (fixed systems such as vibration–strain–displacement-based sensorial data collection [24,25,26] and possibly NCT) are used for identifying the dynamic behavior of the whole structure. An NDT/E consists of more sophisticated tools to detect specific local damages inside the structure, e.g., by using ultrasound, ground penetration radar, infrared cameras, and electromagnetic methods [27,28,29,30,31]. Additionally, NCT, such as computer vision tools including RGB and IR cameras and light detection and ranging (LiDAR), have demonstrated their efficiency in many applications both at the local level (e.g., crack or spalling detection) and global level (e.g., displacement and vibration monitoring) [4,32,33]. The condition assessment methods are illustrated in a diagram in Figure 1.

Recently, new technologies have been experimented with to be employed in the SHM field, and accordingly the application levels of SHM can be presented from another angle (Figure 2). As researchers’ attention has turned to robotic applications, they have started performing robot-based experiments [34,35,36] recently. These robot-based applications can be used at both global and local levels in the continuum of SHM, as presented in Figure 2. For instance, an unmanned ground vehicle (UGV) can employ the vision sensors installed into its platform to collect acceleration data for modal identification (global SHM) and simultaneously implement image-based damage detection (local SHM). Furthermore, because the acceleration data acquisition via accelerometers installed on an UGV increases the spatial resolution of the data, the local defects in the structure can also be detected (local SHM). An unmanned aerial vehicle (UAV) can be used for image-based damage detection applications (local SHM) and displacement or acceleration measurement (global SHM) through vision sensors equipped on its platform by tracking the chosen features in the captured images [37]. Additionally, while the NDT/E is used to perform local SHM by examining the inside of the structural component, the fixed systems (e.g., accelerometers, strain gauges, potentiometers) may only carry out global SHM applications. NCT, on the other hand, are found in more widespread use in both the global- and local-level SHM continuum. As such, while LiDAR, RGB, and IR cameras can be used on robots, they can also be used as fixed systems or NDT tools.

While the traditional and visual inspection techniques are implemented for most civil structures, SHM methods are applied for some large-sized civil systems—mostly large infrastructure systems—to assess the condition of their structural safety and integrity. Typically, after the service life of a structure, the structure undergoes an assessment regulated by the local and federal codes. The assessment may include SHM sensorial systems, NDT, traditional methods, or a combination of these. Then, the collected data are passed to the data workflow for analysis and evaluation to support the decision-making process about the life span of the civil structure by the experts and stakeholders. The final decision determines whether the structure returns to operational service, needs repairs before returning to service, or is unable to be in service any longer and thus is demolished and recycled. Figure 3 illustrates a typical idealized condition assessment procedure for civil structures.

On another note, while data analysis and evaluation via NDT/E, NCT, robotics, fixed systems, etc., are important, the efficiency of the condition assessment procedure can be achieved with effective data visualization techniques for the following data interpretation and decision-making steps. Therefore, it is critical to enhancing the assessment procedures’ efficiency via effective data visualization methods. In this regard, employing immersive visualization tools such as Virtual Reality (VR), Augmented Reality (AR), and Mixed Reality (MR) can be very advantageous.

### 1.2. Motivation, Objective and Scope

As discussed, there has been much research and development of VR, AR, and MR applications in different disciplines and industries. The AEC industry has significantly benefitted from the use of VR, AR, and MR. Accordingly, the applications of VR, AR, and MR in AEC have become the primary research focus in the literature, and many review studies have pointed out the increase in the number of published studies and review papers in the last few years. It is observed in the literature that while most VR, AR, and MR studies in the AEC industry are related to training and education, heritage preservation, and construction activities, some other studies point out the use of VR, AR, and MR for condition assessment of civil engineering structures. After a thorough literature review of VR, AR, and MR in the AEC industry, it is, however, also observed that no review or discussion study has been presented about their use for condition assessment of civil structures. A total of 25 literature review studies were identified that consider VR, AR, and MR separately [38,39,40,41,42,43,44,45,46,47,48,49,50,51,52], or VR and AR together [53,54,55,56,57,58,59,60], or VR, AR, and MR all together (which is XR) [61,62] in the AEC industry. Only two of these studies consider XR in their review, where they studied BIM in AEC.

Furthermore, while some other studies only target the construction industry, others consider architecture and planning. Moreover, several other review papers study education- and training-related subjects. The number of identified review studies in the literature is plotted over the years and shown in Figure 4. Although there are numerous studies available in the literature on structural condition assessment of civil structures, no review or discussion article is available on the use of VR, AR, and MR or XR for condition assessment of civil structures.

In this paper, it is the authors’ intention to review and analyze the existing studies in the literature on using VR, AR, and MR technologies for the condition assessment of civil structures. To that end, first, VR, AR, and MR are explained in the next section to introduce them to the readers. Then, the methodology of the literature review is presented. Then, the paper continues by presenting the studies conducted on using VR, AR, and MR for the condition assessment of civil structures. However, the studies observed in the literature using MR are presented in this paper under the name of AR due to the common confusion about the fundamentals of AR and MR in the literature (VR, AR, and MR are discussed in the next section in detail). Finally, a discussion is presented on this literature review paper, including the current and future trends and recommendations for the future, followed by concluding remarks.

## 2. Virtual Reality, Augmented Reality, and Mixed Reality

The term Extended Reality (XR) includes VR, AR, and MR, and the areas interpolated among them, which is a set of combinations of different spectrums of the virtual world and real world as introduced by Paul Milgram [63]. In Figure 5, the reality–virtuality spectrum of XR is shown. As described in the figure, while VR is an entirely immersive experience surrounded by a digital environment which isolates the user from the real world, AR enhances the real world with a digital augmentation overlaid on it. MR is a blend of the real and virtual worlds, providing interaction between real and digital elements. In essence, MR involves the merging of real and virtual environments somewhere through the mixed reality spectrum (or virtuality–reality spectrum) (Figure 5), which connects completely real environments to completely virtual ones [64]. This is, however, quite different in VR, which isolates the user from the real world, and the user can interact with the digital elements in the virtual world.

*Virtual Reality (VR)*: As described previously, VR provides a completely digital environment where users experience full immersion. The applications that block the user’s view to provide a fully immersive digital experience are named Virtual Reality. At present, a head-mounted display (HMD) or multi-projected environments (e.g., specially designed rooms with multiple large screens: CAVE [65]) are the most common tools to immerse users in virtual spaces. The VR system allows the user to interact with virtual features or items in the virtual world. The main point of VR is the interactive, real-time nature of the medium provided to the user. Generally, while some VR systems incorporate auditory and visual feedback, some also enable different kinds of sensory and force feedback via haptic technology. Widely used HMDs are the Meta Oculus series, HTC Vive, Samsung Gear VR, and Google Cardboard. The VR applications are primarily used in education and training, gaming, marketing, travel, and AEC industries.

*Augmented Reality (AR)*: AR is a view of the real world with an overlay of digital elements where the real world is enhanced with digital objects. The applications that overlay holograms or graphics in the real world (physical world) are named AR. While the interaction with virtual superimposed content in an AR setting can also be made possible, AR applications are generally used with the aim of enhancing the real-world environment with digital objects, usually providing the user with no-to-minimal interaction with the virtual superimposed content in the AR setting. Currently, AR systems use devices such as HoloLens, Magic Leap, Google glass, Vuzix smart glasses, smartphones, tablets, cameras, etc. While some of the HMDs, e.g., HoloLens and Magic Leap, are used for AR applications, they are also used for MR applications. Similar to VR, AR has a wide range of uses, such as gaming, manufacturing, maintenance and repair, education and training, and the AEC industries.

*Mixed Reality (MR)*: The experiences that transition along the virtuality–reality spectrum (mixed reality spectrum) are named Mixed Reality. MR is also called *Hybrid Reality*, as it merges real and virtual worlds [66]. Presently, MR systems use HMDs for MR applications. These HMDs can be classified into two: HMD see-through glasses (e.g., HoloLens, Magic Leap, Samsung Odyssey+, and Vuzix smart glasses, which enable one to see surroundings clearly with additional digital holograms displayed in the glass that allows interaction with real surroundings) and immersive HMDs (e.g., Oculus Quest 2 and HTC Vive; although they have non-translucent displays which completely block out the real world, they use cameras for tracking the outer world). Similar to VR and AR, MR also has use cases in gaming, education and training, manufacturing, maintenance and repair, and AEC industries.

*Side note*: Distinguishing AR and MR applications from each other can be quite challenging due to a common confusion about the fundamentals of AR and MR (as also observed in the literature). Therefore, the studies that use MR in the literature are presented together with AR in Section 4.2 (AR for Structural Condition Assessment) in this study.

Lastly, game engines and other programs are commonly used to develop VR, AR, and MR applications. Unity and Unreal engines are the most commonly used platforms for application development. Vuforia, Amazon Sumerian, and CRYENGINE are some of the others that are also utilized for developing VR, AR, and MR applications. The developed applications are then integrated into the standalone or PC-connected HMDs or other types of devices, mostly operating on Android, iOS, or Microsoft Windows, to be used by the end-user.

## 3. Research Methodology

The research methodology followed in this paper was implemented in two phases (Figure 6). First, the Exploratory Review consisted of an initial literature search, followed by data analysis. Based on the findings of the Exploratory Review, the authors continued with the Refined Review, where more precise data collection and data analysis were carried out, followed by a discussion of the study and current and future trends. In the Exploratory Review and Refined Review phases, Scopus and Google Scholar were used, which are the two most popular paper database search engines. The keywords used in the data collection of the Exploratory Review consisted of more general terms and addressed broader subjects in the civil engineering field. The keywords used in the data collection of the Refined Review used on-target keywords, which were picked based on the results of the Exploratory Review. In the Exploratory Review phase, general keywords were used in both Scopus and Google Scholar where “X” denotes the word “AND”, which connects the keywords together, e.g., Civil Engineering AND Virtual Reality (Table 1). In the Refined Review phase, on-target keywords were used in both Scopus and Google Scholar in a similar fashion to Table 1 (shown in Table 2).

Furthermore, the number of studies included in the literature review is shown in Figure 7, where the PRISMA flow diagram illustrates the study identification process [67]. During the study identification process, a total of 334 studies were obtained. Then, the obtained studies were screened to remove duplicate, nonrelevant, and inaccessible records. The remaining number of studies was 43. Thus, this paper reviews 43 studies: 16 VR and 29 AR (including MR) studies conducted on the condition assessment of civil structures.

## 4. Extended Reality for Structural Condition Assessment of Civil Structures

There are various use cases of VR, AR, and MR in the AEC industry and they are shown in Figure 8. Generally, the use cases of VR, AR, and MR in the AEC industry can be classified as follows: heritage preservation [68,69,70]; design, analysis, and modeling [71,72,73,74]; training and education [43,75,76,77]; construction activities [78,79,80,81,82]; others (e.g., building energy efficiency, emergency and disaster management) [60,83,84,85]; and structural condition assessment (studies are presented in the following subsections). Some use cases have more defined boundaries as they [43,75,76,77] do not cross into other use cases such as training and education. Conversely, some others overlap with other use cases such that the application purposes of heritage preservation and structural condition assessment can be very similar since one of the concerns both use cases share is “safety”. The use case of heritage preservation concerns the safety and preservation of ancient structures. Similarly, the use case of structural condition assessment concerns the safety of existing civil structures. The use case of heritage preservation concerns not only civil structures, e.g., buildings, monuments, and roads, but also artifacts, works of art, folklore, traditions, etc., whereas the use case of structural condition assessment focuses on civil structures and maintaining their safety and structural integrity. Some of the studies that address heritage preservation concern the safety of old civil structures and employ any XR tool included in the use case of structural condition assessment in this paper.

The number of studies included in this paper for VR and AR in the condition assessment of civil structures is plotted over the years (Figure 9). The increase in the number of studies in the last five years is noteworthy and can be linked to the developments in technology. It is also noteworthy that the AR-related studies for condition assessment of civil structures appeared starting from the 2000s. Although there are earlier studies observed in the literature addressing structural condition assessment using VR [86,87,88,89,90,91], those studies display the virtual environment on digital screens, e.g., monitors, tablets, etc., without providing the user with high immersion, unlike HMD or CAVE-like systems. Therefore, this paper does not consider the VR-related studies that do not use immersive systems (e.g., HMD or CAVE-like systems). The following subsections introduce the reviewed studies in this paper in the order of their online publishing date.

### 4.1. Virtual Reality for Structural Condition Assessment

#### 4.1.1. 2017 (One Paper)

The first study, presented in 2017 [92], addressed the challenge of inefficient communication during structural inspections among project stakeholders, engineers, inspectors, and others in the AEC industry. The authors pointed out the importance of a collaborative work environment. Therefore, the study introduced Collaborative Virtual Reality (CoVR), which provides an immersive collaborative platform in a VR environment to facilitate communication between different parties. The developed CoVR is used to test a building inspection experiment with 71 human users. Based on the user experience results, the authors indicated that the CoVR improved communication between parties. They also noticed that users performed better inspection tasks using CoVR than using a single-person VR system.

#### 4.1.2. 2018 (Three Papers)

The same authors of the previously explained study [92] introduced another study [93] where the level of sufficiency of communication between the parties during the structural inspections was investigated. The authors stated a hypothesis that excessive levels of communication could cause information overload and group polarization between the parties, which compromises the performance of the building inspection process. Thus, the authors tested this hypothesis on 71 human users with groups of users and a single user who performed the inspection process in a developed VR environment. Based on the indicators used in the study, the authors concluded that excessive communication negatively affects building inspection performance.

Omer et al. [94] pointed out the difficulties faced by engineers and inspectors during bridge inspections and introduced a bridge inspection workflow to be carried out in a VR environment. A 3D model of the bridge structure (consisting of cracks, corrosion, spalling, etc.) was formed based on the LiDAR capture and then integrated into the VR environment. A MATLAB in-built script was also used on LiDAR outputs to detect defects. The proposed workflow is very promising for efficient bridge inspection. As this study is one of the first in the literature, more experiments and evaluations must be carried out on the bridge inspection workflow in the VR environment.

In [95], the authors addressed the challenges of conventional human-performed visual inspections. Therefore, the authors proposed a mobile robotic system, which is equipped with multiple cameras to capture images of the surface walls of a tunnel structure for a more efficient visual inspection system. Subsequently, the images are reconstructed to create the photogrammetric model, which is then visualized in a VR environment. The authors concluded that the developed methodology in the study could be very beneficial for wall surface documentation, remote inspection, and analysis. This study has some gaps to be filled in the future, such as testing the introduced framework in various other civil structures.

#### 4.1.3. 2019 (Four Papers)

The same authors of [94] published a similar study [96] where a 3D model of the bridge structure was reconstructed based on the LiDAR outputs and integrated into the VR environment (Figure 10). This study specifically emphasized the differences between structural inspection procedures in a VR environment and traditional inspection methods. The authors concluded that the inspection in VR promises to be highly efficient regarding the interpretation of results, accessibility to critical locations, and safety of inspectors. While this study can be very advantageous for bridge inspections, future studies should incorporate multiple user features, such as having more users collaborating in the same VR environment.

In another study [97], the authors proposed an approach where SHM is integrated into a VR environment with the aim of cultural heritage management. Essentially, the study first developed 3D models of a historical church structure using terrestrial and aerial photogrammetry. Then, the models were processed to be integrated into the VR environment. Furthermore, accelerometer sensors were also installed in the structure to display the structural state data of the church structure in the VR environment, such as the natural frequencies, damping coefficients, and maximum and RMS accelerations that the users in the VR environment can observe. This study has good potential for improving touristic, educational, and research activities.

Lee and Ahn [98] proposed a VR-based remote diagnosis framework where the visual and geospatial information of a structure is displayed in the VR platform to assist the maintenance technicians in diagnostics remotely. The study aimed to diminish the time and cost of field visits by using the VR-based remote diagnosis framework. The framework was tested with maintenance technicians, and overall they were observed to be satisfied except for the navigating and viewing information in the VR platform, which caused dizziness.

In [99], the authors addressed the importance of gestural input in a VR environment during the structural inspection procedures of a displayed photogrammetric 3D model. Therefore, the authors proposed PhotoTwinVR, an immersive gesture-controlled system for inspection procedures of 3D photogrammetric models in VR, allowing users to perform basic engineering operations on the model. PhotoTwinVR was tested with three domain-expert participants. The authors concluded that the test results demonstrated the potential of the proposed approach to be applied in practical, real-world scenarios. While this study investigated gestural inputs for inspection procedures, a critical matter for VR-based inspections, there is still room for experimenting with the methodology with more experts.

#### 4.1.4. 2020 (One Paper)

In [100], the authors emphasized the importance of preserving historic structures with the help of VR technology. The authors stated that an effective monitoring system for these structures is essential. Therefore, the authors proposed an IoT-based remote SHM system, which was integrated into a VR environment (Figure 11). While the SHM sensors displayed the environmental and structural data collected from the structure in VR, the users could also interact with the photogrammetric 3D model of the structure whose images were taken with UAV. Additionally, a crack detection application via UAV was also proposed with the help of manually placed targets which helps inspectors promptly inspect the structural damages. The authors concluded that the study could open new scenarios to support SHM activities. Although this study is important for heritage conservation, the manual target placement for crack detection applications may be cumbersome.

#### 4.1.5. 2021 (Two Papers)

In [101], the authors explored gestural input for engineering surveys of civil structures in VR. The study addressed that the effective methods for interacting with photogrammetric 3D models in VR are underexplored in the literature. Therefore, the paper conducted a qualitative case study asking six domain experts to perform engineering measurement tasks in the VR environment. The case study indicated that gaze-supported bimanual interaction of 3D models is a promising method for experts (Figure 12).

The same authors of [96] conducted a complementary study [102] where they compared the conventional visual inspection approach with a visual inspection in a VR environment that consisted of a LiDAR 3D model of the reinforced concrete highway bridge. The authors observed significant benefits of the inspection procedure conducted in VR and improvements over the conventional inspection technique. This study has room to improve the comparative analysis, such as providing insightful indices about the pros and cons of both conventional and VR-based approaches.

#### 4.1.6. 2022 (Five Papers)

Luleci et al. [12] addressed the challenge of site visits and inspections during the SHM procedure, which can be timely, costly, and risky. The authors proposed an approach where a civil structure (footbridge) and its operational SHM data are fused in a collaborative VR environment. For that, tablet LiDAR, terrestrial LiDAR, and UAV-based photogrammetry methods were used to produce varying forms of the 3D model. Moreover, acceleration data was collected from the footbridge to perform modal identification and time history analysis in Finite Element Analysis (FEA) software for SHM data. Then, these 3D models and SHM data were fused in the VR environment where multiple users could visualize, analyze, and interpret the bridge behavior (Figure 13). The authors concluded that the collaborative VR environment proposed in the study offered very beneficial features to optimize the number of site visits during inspections and SHM applications. While this study is important for collaborative inspections and remote diagnosis, a substantial effort is needed to establish such a VR environment.

The authors in [103] pointed out the recent increase in robotic applications in structural inspections. They indicated that a more profound insight must be gained to enable seamless human and robot collaboration during building inspections. Thus, the authors conducted a comparison study of a robot- and human-based inspection process. In the study, while the robot is simulated in ROS (Robot Operating System), which scans the building, the human conducts the same inspection task in a VR environment. Based on the total identified structural defects, the path that was followed, and the total time used, the human-based inspection demonstrated better performance in the VR environment. One of this study’s gaps is implementing the methodology in a real environment instead of a simulation using ROS.

In the next study [104], the authors addressed the importance of inspections in construction projects. The paper investigated the collaboration of VR and robotics for remote inspection (although this study was conducted for construction quality and progress, it can also be very beneficial for the condition assessment applications). In the study, a quadruped robot in the real environment was commanded by a human inspector through a VR controller in real time (Figure 14). Based on the investigation, the authors listed the five benefits and challenges of using robots through VR in construction inspection and monitoring where the top benefit was enhanced collaboration and the top challenge was low-resolution display. Although this study is significant for remote inspection procedures, incorporating a collaborative environment where multiple users could join would benefit the study significantly.

In another study, Savini et al. [105] investigated the integration of current technologies into inspection and SHM applications for the purpose of heritage conservation of infrastructures. A VR environment was developed that consisted of a CAD and photogrammetric model of a historical bridge structure (Figure 15). Additionally, to integrate the health monitoring of the bridge, acceleration data were collected from the structure and linked to the bridge in the VR environment to enable the visualization of the data and metadata to the users. The authors also discussed the study’s advantages and challenges in using such a VR information system. This study is critical because it demonstrates the feasibility of integrating CAD and photogrammetric models in a VR environment for heritage conservation purposes.

In [106], the authors proposed a VR-based methodology for developing an autonomous structural health inspection system via UAV. In the study, the authors experimented with the methodology in the built environment with a virtual UAV utilizing an open-loop approach in the VR environment, including SLAM (Simultaneous Localization and Mapping) and waypoint-based control. The authors also explored the potential of aerial robotics for developing modern data-driven structural space exploration, damage assessment, and optimal control.

### 4.2. Augmented Reality for Structural Condition Assessment

#### 4.2.1. 2000 (One Paper)

One of the first studies in the literature aimed at improving [107] construction, inspection, and renovation methods for building structures. The developed AR system generates virtual computer-made objects to the surrounding physical environment in which the objects are overlaid in the real environment. For instance, as the person who wears the HMD moves their head toward the structural columns in the building, the location of the rebars and structural analysis of the column can be visualized in the HMD.

#### 4.2.2. 2008 (One Paper)

In another study [108], the authors conducted an experiment to evaluate the benefits of structural inspection with an AR prototype system (ARCam) over a conventional technique. The study tested the ARCam on a steel column along with the conventional method. The test results indicated that although the AR method was less accurate than the conventional methods, it could still meet standard tolerances. With further advancements in AR technology, the authors concluded that ARCam is a promising inspection tool for inspecting steel columns.

#### 4.2.3. 2012 (One Paper)

Dong et al. [109] addressed the difficulty of the buildings’ Interstory Drift Ratio (IDR), which is critical for catastrophic events that determine the structure’s safety for occupancy. Thus, the authors introduced using AR technology with a non-contact method for determining the IDR. The method was tested with an AR-based HMD in a virtual prototyping environment (Figure 16). In essence, the methodology identified the corner locations along with the vertical edges of buildings for defining the IDR. Additionally, the authors conducted a sensitivity analysis of the introduced methodology for potential use in practice. Although this study is innovative and demonstrated success, the HMD they designed and built has a complex mechanism and may not be practical to use in the field.

#### 4.2.4. 2014 (One Paper)

The authors in this paper [110] proposed a framework that implements an inspection of building structures by remotely controlling an AR.Drone via Google glass as an HMD. The major benefit of using Google glass as the navigator (controller) for visualization is the synchronization of the relative rotation of the user’s head (who is wearing the Google glass) with the AR.Drone’s movement, as well as gestures performed by the user. The framework was tested on a building structure, and the authors discussed possible methodological improvements, such as integrating a direct connection between Google glass and AR.Drone, which is believed to enhance the quality of network transmission for communication.

#### 4.2.5. 2017 (Two Papers)

In another study [111], the authors proposed an AR method for measuring segment displacement during tunneling construction. The author’s aim was for the AR model to provide the baseline established based on the quality standards in which the AR model is overlaid on the real structure to enable the measurement of segment displacement. Thus, structural safety can be sustained automatically by measuring the differences between the baseline provided by the AR model and the real-world view. The authors tested a prototype AR model on a real tunnel structure. The results indicated that site inspections can be conducted effectively and at a very low cost with the proposed AR model compared with the conventional method. While the framework was tested on a tunnel structure, subsequent studies should experiment with different civil structures such as bridges.

Fonnet et al. [112] addressed the difficulties in the renovation decisions of heritage buildings. The authors emphasized the importance of BIM models for heritage buildings and introduced hBIM (Heritage BIM). To overcome the difficulties in inspection data collection and integration into the BIM model, the study proposed using HoloLens. To document the data acquired during the inspections of the heritage buildings, the authors recommended using HoloLens to replace hand tools and cameras for visual inspection. The study also investigated the advantages and disadvantages of HoloLens, such as user-friendliness. They recommended initial training for inspectors to familiarize them with the device. Some hardware disadvantages, such as low battery life and a narrow field view, were also stated. The evaluation was not limited to the general use of headsets. The authors concluded that HoloLens provides a great advantage for structural inspections; however, some technological components need improvement to be widely used among inspectors.

#### 4.2.6. 2018 (One Paper)

Schickert et al. [113] implemented a method that enables NDT data to be used in digital building models under the Building Information Modeling (BIM) framework. Thus, an AR application was developed which overlays the tablet’s camera image of a concrete specimen with ultrasonic and radar images of the specimen’s interior (actual data) and the 3D planned geometric geometry of the built-in parts (target data). As a result, the geometric relation between the camera and the specimen’s inner image is preserved in the case of the tablet’s movement or rotation. The study’s goal was to directly allow model-based inspection and maintenance chores on the real structure by supplying additional planning data and measurement results in the structure’s real environment.

#### 4.2.7. 2019 (Six Papers)

In the following study [114], the authors pointed out the difficulty of data documentation, access, and visualization during the inspection and SHM applications. Therefore, they aimed to develop a method for collecting, integrating, accessing, and visualizing the data and metadata of SHM suitable for both on- and off-site use. The method was implemented with a tablet using an integrated AR kit development tool to enable informational modeling, e.g., visualizing the sensorial data information through the tablet’s camera on the tablet’s screen. The authors concluded that the method would enable more efficient on- and off-site presentation of engineering assessments and foster communication between different parties. This study is essential for SHM applications and could be improved further with real-time sensorial data information access.

The authors of [115] pointed out the difficulty of collaboration during the inspection process as every inspector documents the inspection records separately in paper format. Therefore, the study introduced an AR approach to assist the inspectors during the inspection process. The proposed AR-based approach uses a tablet that enables inspectors to geolocate, annotate, and make other informative edits on the captured defects during the inspection, which can be simultaneously seen and modified by the other inspectors (Figure 17). This work holds significant importance for collaborative inspection procedures performed by inspectors at the same time.

Karaaslan et al. [11] introduced a smart mixed-reality framework for inspecting concrete bridges. The study proposed to use HoloLens to facilitate human–AI interaction. Rather than an AI completely replacing the human, the AI-based inspection eliminates the subjectivity of visual inspection while benefiting from human expertise using a mixed-reality platform. The authors also deployed a structural defect identification module in HoloLens to assist the inspectors with the defect identification process. The introduced framework with HoloLens also allows inspectors to measure the identified defect sizes (Figure 18). Although the framework introduced is innovative, it is important to ensure that using HoloLens is feasible for inspectors for the bridge inspection process.

Brito et al. [116] addressed the significance of preserving and inspecting heritage buildings. Thus, they employed an approach for easing the maintenance tasks of inspectors by using HoloLens. The proposed approach has several features, such as overlaying the inspected model on the real world in the HoloLens, structural damage report filling, data storing, and annotating pictures and audio of the reports. This work is significant for inspection procedures and the assessment documentation process by the inspectors.

In another study [117], the authors introduced a methodology to assist the inspectors with crack identification tasks in which the introduced framework automatically verifies the identified cracks via HoloLens, deployed with a crack detection algorithm. Essentially, HoloLens assists the inspector by first sensing the 3D surroundings, then identifying the cracks from the captured images, and then displaying the identified cracks on the real structure where it is displayed on the HoloLens’s glasses to the inspector. The inspector can also manipulate the crack data with simple hand gestures. Based on the experiments, the authors confirm that the inspectors can accurately acquire the crack data’s presence, location, and size by only using the HoloLens glasses.

Dang and Shim [118] addressed the challenge of storing the damage and repair records and the in situ structural behavior of bridge structures during their inspection. Thus, the paper proposes a real-time Bridge Management System (BMS) that employs a BIM approach in cooperation with a HoloLens device to automate inspection tasks. The proposed method is applied on an existing cable-supported bridge and demonstrates good potential for enhancing the performance of maintenance activities. Implementing BMS for BIM models with HoloLens is a very innovative approach, and this is one of the first studies presenting it in the literature.

#### 4.2.8. 2020 (Seven Papers)

In another study [119], the authors investigated the benefits of AR for infrastructure inspection procedures and providing erroneous data to the users or inspectors. In this study, the authors examined the impact of AR cues throughout varying degrees of target saliency to measure the overall performance in a signal detection task. The authors experimented with 28 participants, in a virtual environment of a bridge, who flew a UAV for the signal detection task. Results indicated considerable variations in the false alarms within different target salience cases; however, no huge variations were observed throughout the AR cue types for the hits and misses.

Kilic and Caner [120] addressed the importance of using NDT in collaboration with AR technology for bridge condition assessment. The paper presented the benefits of using AR for visual inspection combined with Ground Penetrating Radar (GPR), Laser Distance Sensors (LDS), Infrared Thermography (IRT), and a Telescopic Camera (TC). The proposed approach was tested on a bridge with cracks and corroded rebars in its structural system. The authors concluded that the proposed methodology could improve the asset owners’ and/or engineers’ decision making based on the results of the AR-assisted NDT approach. Using AR in collaboration with NDT is an up-and-coming framework for bridge condition assessment, and this work demonstrates some aspects of this collaboration.

The authors in [121] addressed the challenges of human-performed inspection processes. The authors proposed an automated damage detection framework in real time using AR glasses where the results from the module of deep learning (DL)-based corrosion/fatigue detection, classification, and segmentation were deployed in an AR HMD. Additionally, a module of damage detection in multi-joint bolted regions was also incorporated into the AR glasses. Then, the authors tested the introduced framework on a bridge structure and obtained promising results (Figure 19).

In another study [122], the authors pointed out that although UAV-based inspection has been widely accepted for building inspection recently, its advantages are not fully used due to the absence of knowledge from the UAV-based inspection to assist the decision-making phase. Thus, the paper introduced an AR-based solution by integrating BIM and UAV. This novel integration enables seamless knowledge transfer from BIM to enhance the video captured by the UAV during the inspection. The authors concluded that according to the test results of the AR system prototype on a building structure, more efficient, extensive, and unbiased UAV-based building inspection is possible.

In the following work [123], the authors studied 3D visualization of the ultrasonic test (NDT) using HTC Vive. The authors calibrated the spatial features of ultrasonic data with the tested specimen, enabling visualization in the HMD (HTC Vive) with the 3D tracking system. The visualization of ultrasonic data in HMD is displayed in real-time, directly in the AR HMD. As the NDT data can be visualized through the HMD, the introduced framework in this study could be beneficial for assisting inspectors during structural inspections (Figure 20). Although this study was implemented on a helicopter propeller, the same approach using NDT could be used for a bridge girder or decks or other structural elements as NDT/E applications can be very advantageous for condition assessment of civil structures [29].

The authors of [124] addressed the various challenges of visual inspection procedures, which causes ambiguity in the structural assessment. Therefore, the authors proposed using HoloLens to improve the ability of inspectors during infrastructure inspections. Using HoloLens enables the inspector to make decisions faster and more accurately, assess risks on the site, measure damage growth, and create inspection documentation to store the data collected during the inspection. The HMD was tested on several concrete bridges and standalone girders for simple inspection tasks, e.g., measuring crack sizes and documenting them. The authors concluded that the proposed approach is promising and ready to be used for infrastructure inspections. This work demonstrates another beneficial application of using HoloLens for inspection procedures.

Maharjan et al. [125] developed and validated a novel human–machine interaction system through HoloLens, which assists the inspector in the field during data collection and decision-making processes. In essence, the study integrated two new applications with the HMD: a sensor–MR connection and a QR code–HoloLens connection. Through these new developments, the authors built a new interface in the HMD for inspection, enabling other developers to build and deploy more AR applications for more effective inspection procedures. Overall, the authors studied the role of AR in increasing inspector awareness during the inspection work.

#### 4.2.9. 2021 (Three Papers)

The authors of this technical report [126] studied using HoloLens for crack detection and measurement on bridge structures to tackle the limitations of human-performed on-site inspection procedures. The crack detection/characterization module the authors developed was deployed in the HoloLens glasses where it assists the inspector (user) with its automated hands-free data collection ability via the HoloLens built-in camera sensors. The authors implemented multiple experimental tests in the laboratory and field where they tested the crack identification application in HoloLens. The authors concluded that using HoloLens has the potential to assist the inspection procedures.

In another study [127], the authors pointed out the challenge of access to sensorial information, such as displacement data in the field, as the sensorial data collected in the field first undergoes additional processing and is shared with sensor and SHM specialists. The authors envisioned that if the inspectors could observe the displacement data of a location of interest in real time during the inspection process, this would allow inspectors to generate a new information-based decision-making reality on the site. Thus, the study built a novel human-centered interface which supplies inspectors with real-time access to sensorial data, e.g., real-time displacement on HoloLens glasses during the inspection. The proposed interface was tested with laboratory experiments and its accuracy was verified with a laser for the displacement data. The authors concluded that with the introduced interface for HoloLens in this study, the inspectors could observe the displacement data, share it, and visualize it in time history (Figure 21). This study is one of the first to visualize SHM sensorial data in HoloLens. It could be very beneficial to employ in the field during data collection.

Xu et al. [128] proposed an application named Time Machine Measure (TMM) on HoloLens 2. The working principle of the TMM application is to bring up the saved meshes of formerly scanned environments and then overlay them onto the real surroundings. Thus, it assists the inspectors in intuitively tracking the changes, e.g., deformation in the structure over time, by measuring the distance between the restored meshes and the meshes of the current real environment. The TMM application proposed on HoloLens 2 was validated with experiments. Although this application demonstrated advantages, one of the major concerns here is the precision of the location of the meshes for measuring deformation.

#### 4.2.10. 2022 (Four Papers)

The authors of [129] introduced a BIM- and AR-based supportive inspection system named BASIS. With BASIS, the inspectors can obtain the necessary information about the bridge model (e.g., historical defect information—defect class, size, and location), and the relevant data are overlaid on the real structure on the tablet’s screen. The prototype of BASIS was tested on a pedestrian bridge, and it was observed that BASIS can help inspectors to acquire accurate inspection data and minimizes the data subjectivity caused by human judgement and/or errors. Although this study demonstrates that it can be helpful for inspectors, it should also be tested on different civil structures.

In [10], the authors explored different real-time machine learning models to be deployed in Mixed Reality headsets to inspect concrete bridges visually. The study focused on models that can perform in real time in small edge devices and also investigated the models’ accuracy, inference speed, and memory size. The study used two machine learning models for defect localization and quantification deployed in the headset. This enables the inspector to interfere with the AI’s performance by using hand gestures and voice commands. The study also recommended a methodology for image registration in HoloLens, which allows the 2D image to be registered in the 3D space of HoloLens.

The authors of [130] presented a BIM-based application where HoloLens is used to improve and facilitate the management of bridge inspections and maintenance procedures from the office. The authors named the application HoloBridge. The study addresses the inefficiency of decision making during inspection and maintenance tasks, which are generally carried out on hard-copy sheets and 2D drawings. The application consists of modules that enable users to examine and update the ongoing inspection and maintenance activities. The application the authors developed was tested on an existing bridge as the case study (Figure 22). This study is very promising for remote bridge condition assessment from the office, but developing this kind of application may take a great effort.

Al-Sabbag et al. reported a study [131] on Human–Machine Collaboration Inspection (HMCI) to allow collaboration between inspectors who wear HMDs and robots for structural inspection procedures. While the inspector can obtain the meta-information about the defect on the site via HoloLens 2, the assistant robot gathers that metadata. It processes it to an off-site computational server in real time. The workflow of HMCI begins with the robot generating the 3D map of the site, and the spatial coordinates are calibrated with HoloLens 2 to be displayed in the HoloLens 2’s view. The produced 3D map and pictures are then sent to the server for damage analysis. Subsequently, the damage results are received by the HoloLens2 and overlaid on the real scene where it is visualized in the HoloLens 2’s view (Figure 23). The proposed study was tested in a laboratory environment. The authors indicated that this study is one of the first on a human–machine collaborative system integrating robots, inspectors, and AR for bridge inspections. This study demonstrates an approach to the structural inspection process, and it is expected to lead to many follow-up studies that improve the concept.

### 4.3. Discussion, Recommendations, and Current and Future Trends

Our review shows that there have been notable research efforts on applying XR for the condition assessment of civil structures. These studies addressed various problems throughout the assessment of civil structures and offered novel solutions to these issues. As data visualization and interaction holds vital importance for condition assessment of civil structures, the utilization of XR technologies can be advantageous for accurate data interpretation for the following decision-making steps. In particular, using XR in cooperation with other emerging technologies such as artificial intelligence (AI) and robotic systems could be a game changer and alleviate the difficulties faced in condition assessment techniques of civil structures, as few studies [11,104,131,132] have presented their frameworks on these. Enabling NDT data and displaying it in real time through HMDs to the users or inspectors could be of great assistance to the inspectors during the inspection procedure as it would be very informative about the defect content hidden in the structural element, which is not possible to observe with the naked eye [113,120,123].

Among the studies presented using VR technology, improving the condition assessment of civil structures by integrating a remote inspection mechanism (off-site inspection) is one of the highlights of the VR studies. Thus, providing a collaborative work environment in VR where inspectors, engineers, and other parties work together on the assessment data results is essential for the decision-making phase. While VR enables a remote assessment mechanism, providing this collaborative work environment also bolsters communication efficiency between the parties. In addition, having such a collaborative work environment and remote field assessment mechanism decreases the number of site visits, lowering the related transportation and other indirect expenses. Furthermore, it is observed that the research focus of the AR studies (including MR—as discussed in Section 2) is to improve inspectors’ efficiency by deploying the HMDs with digital visual aids. While some other AR studies utilize HMD to measure the defect sizes and document the defect data in the central database, other studies take advantage of AI integration in the HMD to automate defect detection intelligently to minimize the inspector’s subjectivity and expedite the inspection process. Moreover, using a robotic vehicle is also found to be an advantageous approach where one of the aims of the robot is to assist the inspection process in the areas inspector cannot reach by sending the data collected via the robot’s sensors to the inspector’s HMD. In conclusion, among the AR studies (including MR), it is observed that they generally aim to reduce human labor in the field. Figure 24 summarizes the highlights observed in XR studies presented in this paper.

*Recommendations:* While immersive visualization techniques such as XR are up-and-coming tools, it is essential to understand what is really needed for the application purpose and how these tools would assist the condition assessment methods used for civil structures. As XR technologies are becoming more accessible, affordable, and mainstream, the research on using them for the condition assessment of civil structures should be on-target. In particular, understanding the benefits of using XR over conventional assessment techniques is vital to their utilization in the field. In doing so, comparing XR over conventional techniques should employ quantification indices for a contextual analogy. As such, these indices should account for, e.g., the accuracy, time taken, and practicality of the technique (such as a comparison of using XR and the conventional techniques utilizing comparable quantification indices). These comparative analysis studies are critical as they indicate comparison results between these techniques, which could expedite the use of XR technologies in practice.

*Current and Future Trends:* Based on the studies summarized in this review paper, one of the current trends for using XR is to reduce site visits by enabling remote field assessment methods through a collaborative work environment where engineers, inspectors, and other parties can join simultaneously. In addition, some other current trends are to reduce human labor in the field and to support inspection activity by providing inspectors with digital visual aids and enabling the interaction of those visual aids with the data observed in the real world. In view of the current trends and the studies reviewed in this paper, more involvement of advanced technological tools in the condition assessment procedures is expected. The technological progress in hardware and software will also enable the use of AI, Robots, XR, and SHM in collaboration with a central unit (human) (human-in-loop approach) for a fully autonomous assessment approach to the civil structures. In Figure 25, the roadmap for condition assessment of civil structures is illustrated where three assessment approaches are identified: Type-I is the type where traditional assessment approaches are taken, e.g., chain dragging, concrete tapping, visual inspection with the aid of heavy equipment, etc. Type-II is the type where sensor-based assessment strategies are used. The assessment workflow relies on the data collection via vision or other types of sensors, whether fixed on the structure or installed on a mobile platform such as a robot. Type-III is the type where autonomous assessment approaches are taken. The aim of Type-III is to automate the condition assessment procedures with a collaboration of Robots, AI, XR, and SHM while the human remains in central control. It is safe to assume that currently we are in between Type-I and Type-II (closer to Type-I). While most civil structures do not undergo a condition assessment cycle, some undergo traditional assessment procedures. Some other large-sized civil structures are assessed via sensor-based techniques. As the research and investigations increase for Type-II and Type-III approaches, their practical applications will be more realizable.

## 5. Summary and Conclusions

The use of immersive visualization tools such as Virtual Reality (VR), Augmented Reality (AR), and Mixed Reality (MR) in the architectural, engineering, and construction (AEC) industry has revealed that these technologies can be very advantageous as they are used for different purposes such as construction activities, training and education, heritage preservation, etc. Civil engineering structures’ health condition (safety) has become critical recently. Effective and practical tools are a pressing need for the condition assessment of civil structures. The use of Extended Reality (XR), an umbrella term for VR, AR, and MR for condition assessment of civil structures, is observed in many studies in the literature. It is also observed that while the available review studies in the literature are presented for different use cases in the AEC industry (Figure 8), the authors of this paper did not identify any review or discussion study about the use of XR for condition assessment of civil engineering structures. A total of 25 literature review studies were identified that consider VR, AR, and MR separately [38,39,40,41,42,43,44,45,46,47,48,49,50,51,52], or VR and AR together [53,54,55,56,57,58,59,60], or VR, AR, and MR all together (which is XR) [61,62] in the AEC industry. Only two of these studies consider XR in their review, where they studied BIM in AEC. Furthermore, while some other studies only target the construction industry, others consider architecture and planning. Moreover, several other review papers study education- and training-related subjects.

Although numerous studies are available in the literature on structural condition assessment of civil structures, no review or discussion study is available on the use of XR for condition assessment of civil structures. Therefore, this paper presented a literature review study on using XR technologies for the condition assessment of civil structures. After a thorough literature review and, subsequently, a record screening process (the process is presented in Section 2), a total of 43 studies were identified and introduced in Section 4. Then, the paper presented a discussion about the studies reviewed, recommendations, and current trends with the outlook for the future. The authors of this paper aimed to provide essential information and guidelines for practitioners and researchers on using XR to maintain the integrity and safety of civil engineering structures.

*Conclusions:* The conclusions of this paper are listed below:It is generally observed from the reviewed studies that the studies use XR to conduct field assessment remotely while simultaneously providing a collaborative work environment for engineers, inspectors, and other third parties. In addition, some other studies use XR to reduce human labor in the field and to support inspection activity by providing inspectors with digital visual aids and enabling the interaction of those visual aids with the data observed in the real world.The first study that used AR for assessment was published in 2000. The first studies on VR, on the other hand, were in 2017. Since 2017, the overall number of studies per year has gradually increased. As XR technologies are becoming more accessible, affordable, and mainstream, more research and development of using them for the condition assessment of civil structures is expected.Understanding the benefits of using XR over conventional assessment techniques is vital to their utilization in the field. These comparative analysis studies are critical as they reveal the comparison results between XR and conventional assessment techniques, which could expedite the use of XR in practice. Therefore, these studies should employ quantification indices for a contextual analogy. As such, the indices should account for, e.g., accuracy, time, and the technique’s practicality.More involvement of technological advancements in condition assessment procedures is expected in the near future. The technological progress in hardware and software will enable the use of AI, Robots, XR, and SHM in collaboration with a central unit (human) for a fully autonomous condition assessment approach to the civil structures.Future studies could perform a comparative analysis of using VR/AR/MR tools, such as different HMDs, for the condition assessment of civil structures. In this regard, each HMD could be listed in terms of its use efficiency for various purposes.

## Figures and Tables

**Figure 1 sensors-22-09560-f001:**
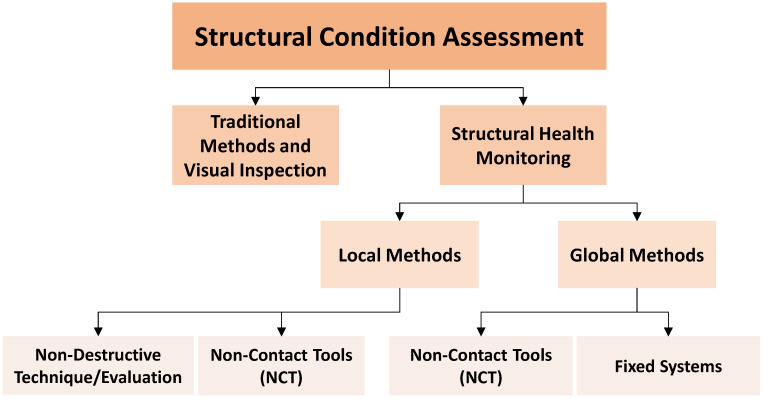
Structural condition assessment methods: traditional methods and visual inspection and structural health monitoring by their application levels.

**Figure 2 sensors-22-09560-f002:**
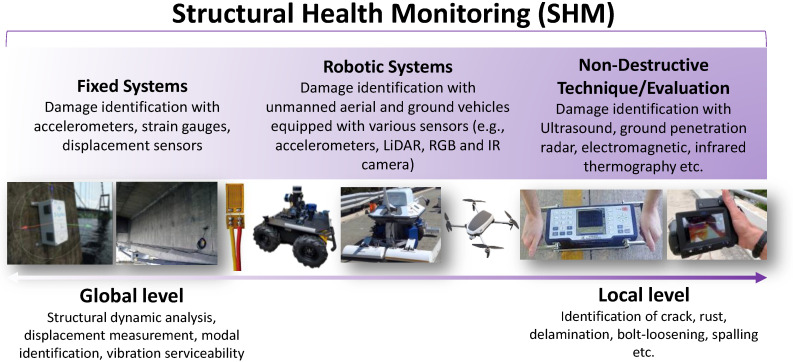
Global- and local-level SHM continuum.

**Figure 3 sensors-22-09560-f003:**
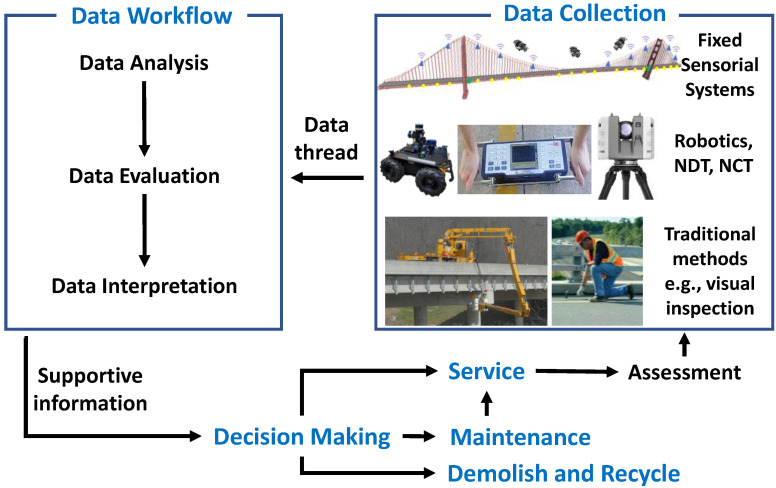
Typical idealized condition assessment procedure for civil structures.

**Figure 4 sensors-22-09560-f004:**
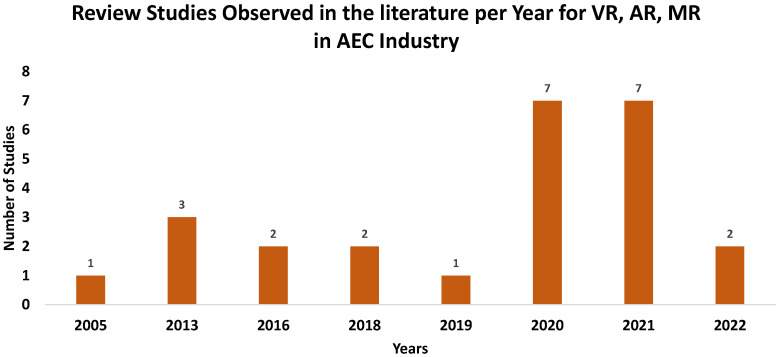
Number of review studies observed in the literature on VR, AR, and MR in condition assessment of civil structures over the years.

**Figure 5 sensors-22-09560-f005:**
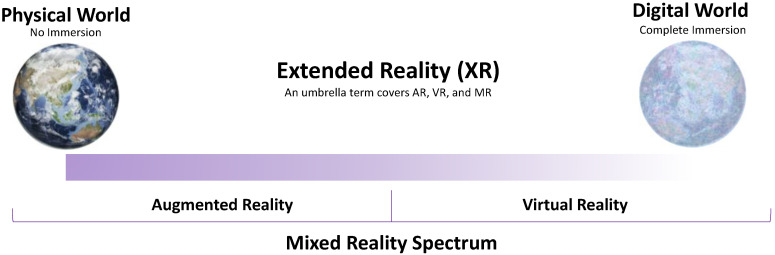
The mixed reality spectrum.

**Figure 6 sensors-22-09560-f006:**
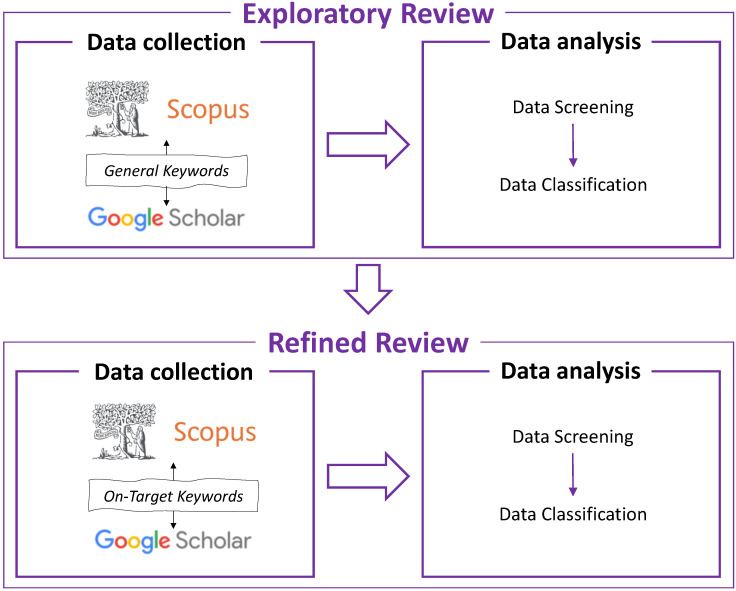
Overview of research methodology.

**Figure 7 sensors-22-09560-f007:**
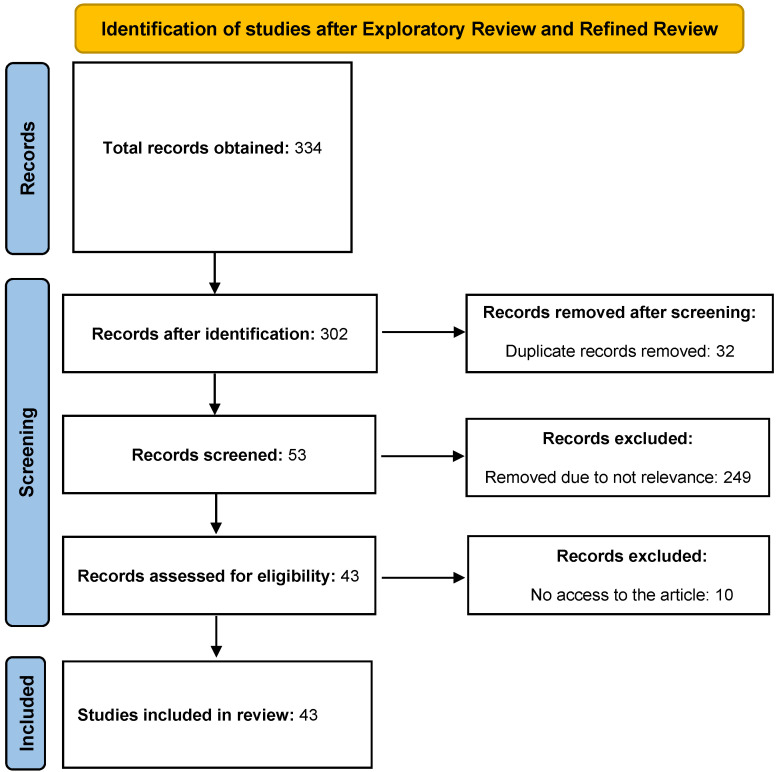
PRISMA flow diagram of record selection according to [67].

**Figure 8 sensors-22-09560-f008:**
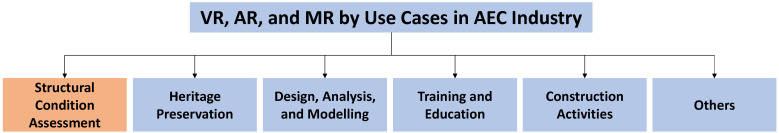
Use cases of VR, AR, and MR in the AEC Industry.

**Figure 9 sensors-22-09560-f009:**
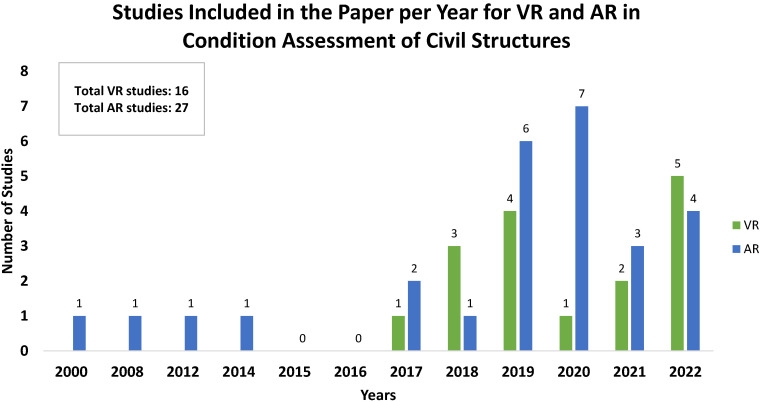
Number of studies included in this paper on VR and AR in condition assessment of civil structures over the years.

**Figure 10 sensors-22-09560-f010:**
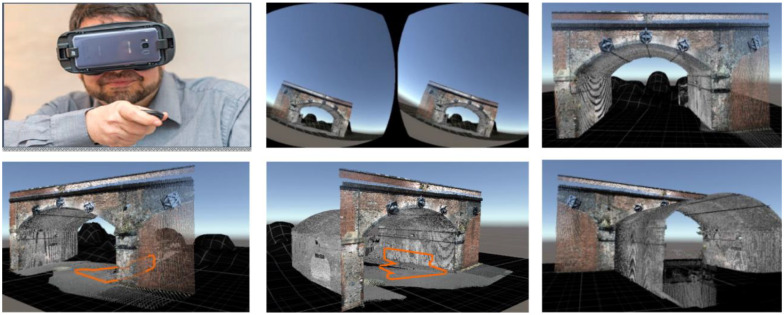
Multiple views of the bridge replicated in a 3D VR environment [94].

**Figure 11 sensors-22-09560-f011:**
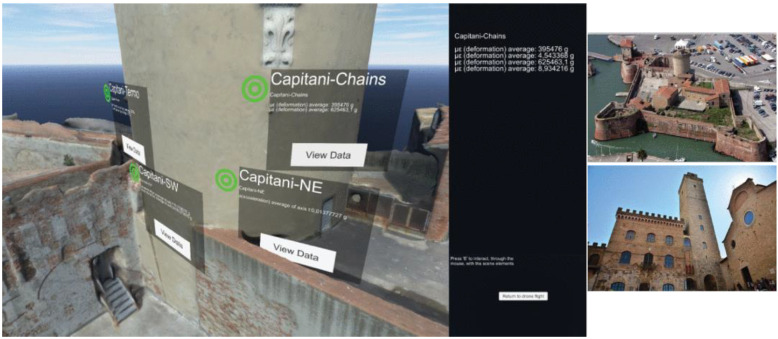
The VR environment: users can explore the virtual reconstruction of a monitored structure while having direct access to values measured by the sensors deployed on it [100].

**Figure 12 sensors-22-09560-f012:**
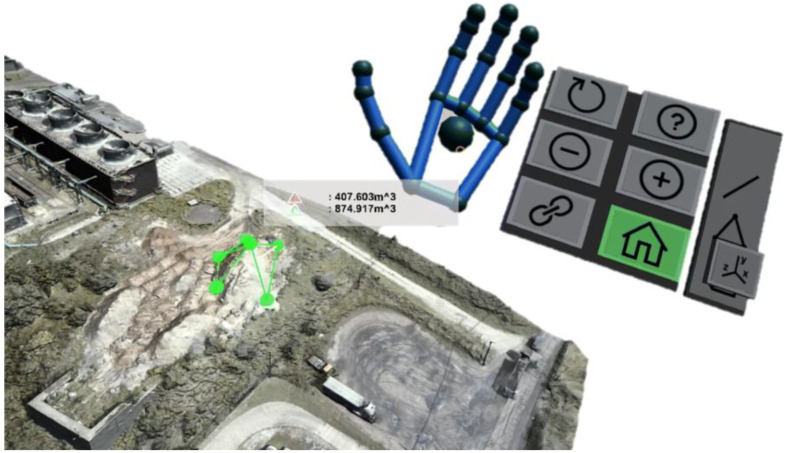
The volume measurement of a pile of sand in the photogrammetry model with hand gestural input [101].

**Figure 13 sensors-22-09560-f013:**
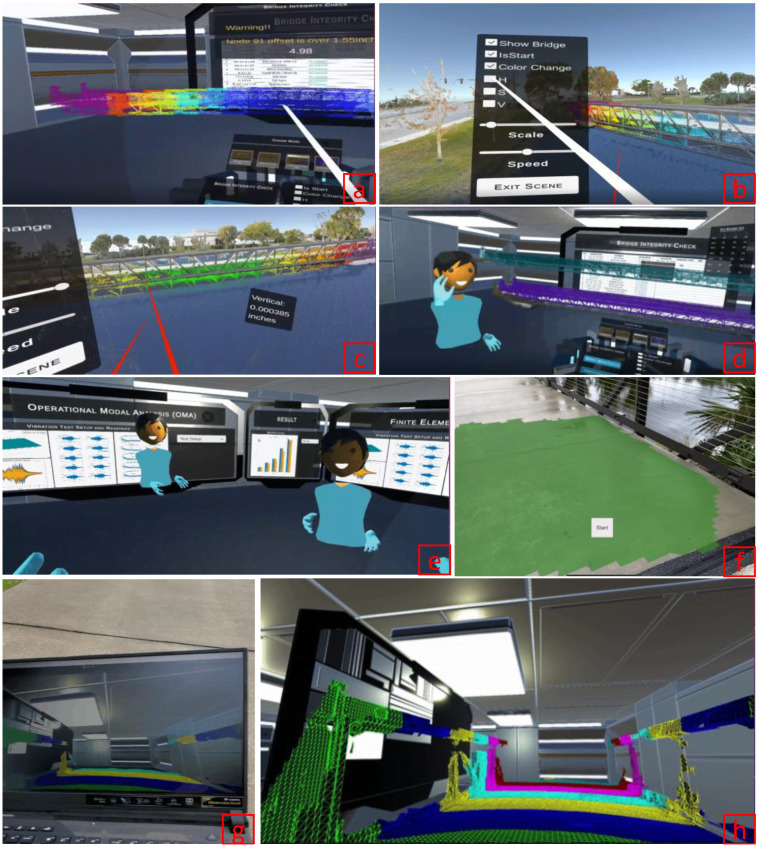
(**a**) FEA reflected TLS point cloud with serviceability limit state check warning and dynamic monitoring of the midspan; (**b**) configuration panel of the FEA reflected TLS point cloud in immersive view; (**c**) dynamic monitoring of the all nodes with VR controller in immersive view; (**d**,**e**) multi-user feature; (**f**) iPad screen—iPad LiDAR footbridge scanning; (**g**,**h**) iPad LiDAR real-time footbridge reconstructing in the VR environment [12].

**Figure 14 sensors-22-09560-f014:**
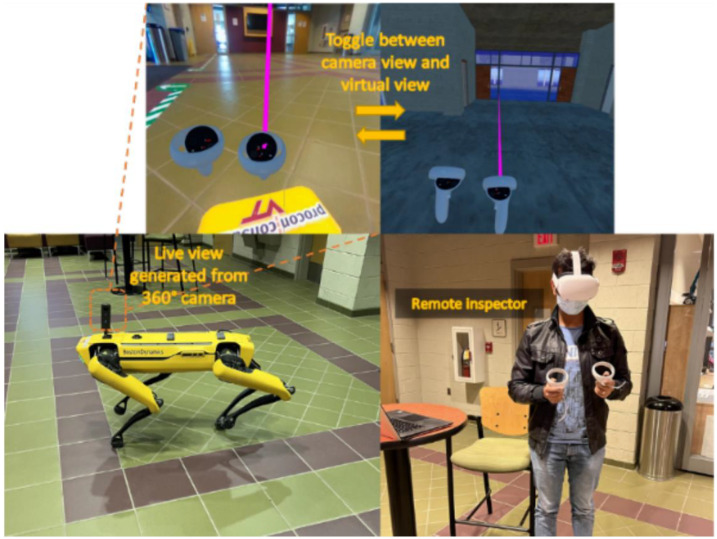
Prototype of inspector assistant robot [104].

**Figure 15 sensors-22-09560-f015:**
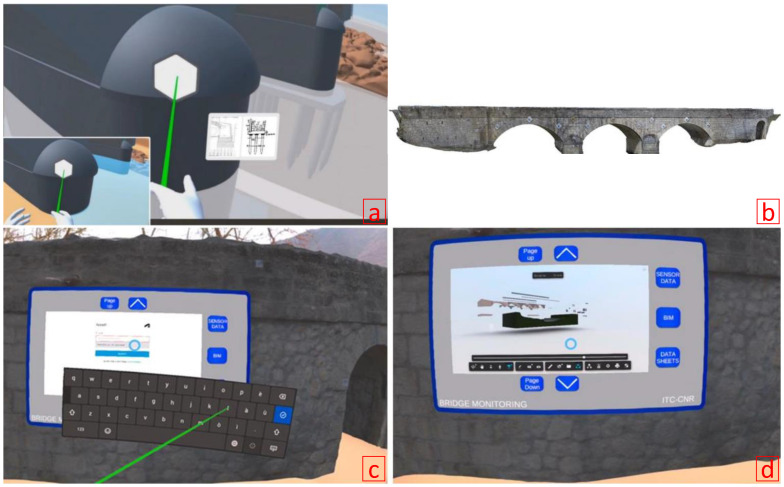
(**a**) VR of non-visible bridge components and related information that clarifies the structure; (**b**) photogrammetry model of the bridge; (**c**,**d**) visualization and query in the VR environment of the BIM model [105].

**Figure 16 sensors-22-09560-f016:**
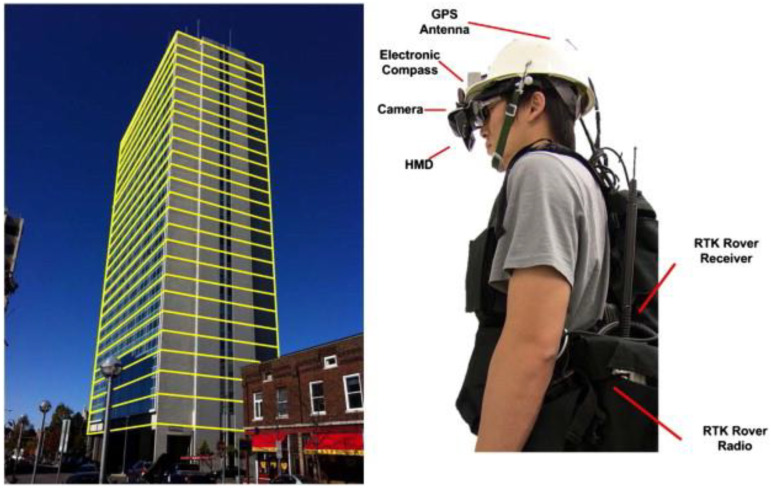
Schematic overview of the introduced AR-assisted assessment framework to estimate the IDR [105].

**Figure 17 sensors-22-09560-f017:**
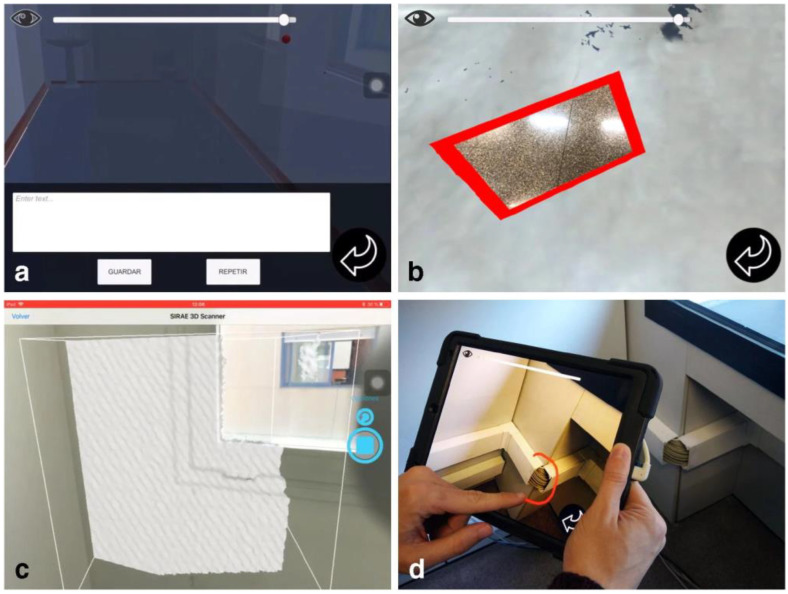
(**a**) User interface for text annotation, (**b**) photographic annotation projection on the floor, (**c**) scanning the corner of a window to build its 3D model, (**d**) the stroke-type annotation [105].

**Figure 18 sensors-22-09560-f018:**
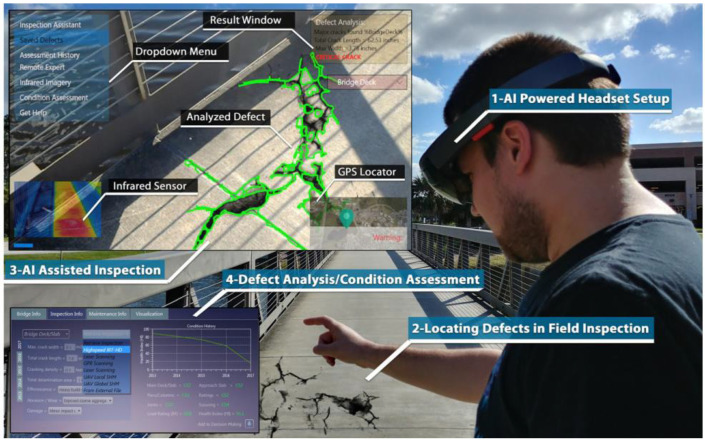
Visual representation of the AI-supported AI framework [11].

**Figure 19 sensors-22-09560-f019:**
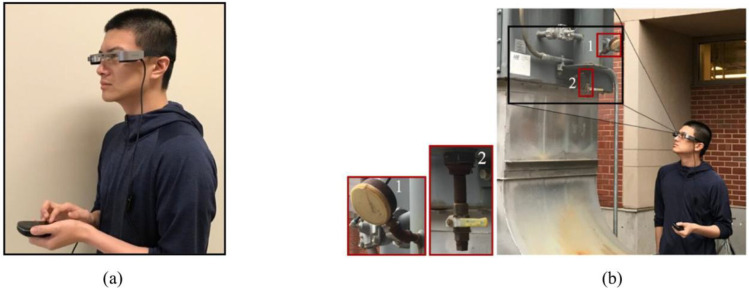
(**a**) Inspector wearing Epson BTB-300, (**b**) site inspection to identify corrosion/fatigue with Epson BTB-300 [121].

**Figure 20 sensors-22-09560-f020:**
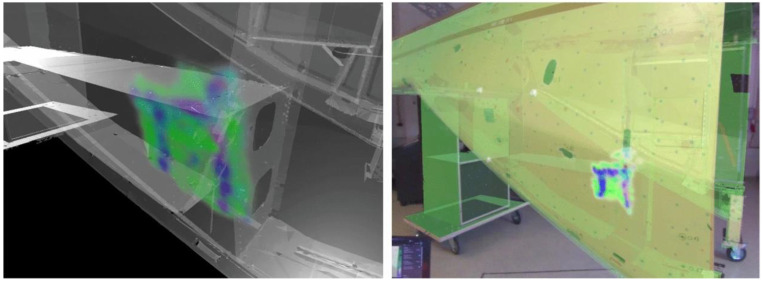
The visualization of the NDT data in VR mode (**left**) and in AR mode (**right**) [123].

**Figure 21 sensors-22-09560-f021:**
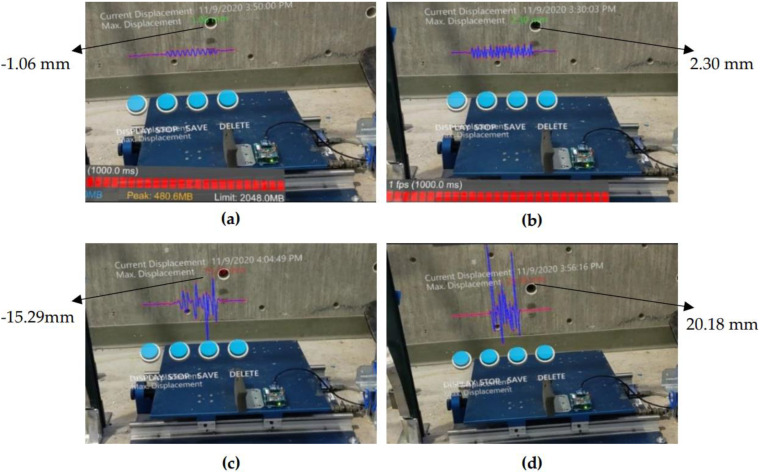
Real-time displacement visualization of different sensors through the HoloLens interface [11].

**Figure 22 sensors-22-09560-f022:**
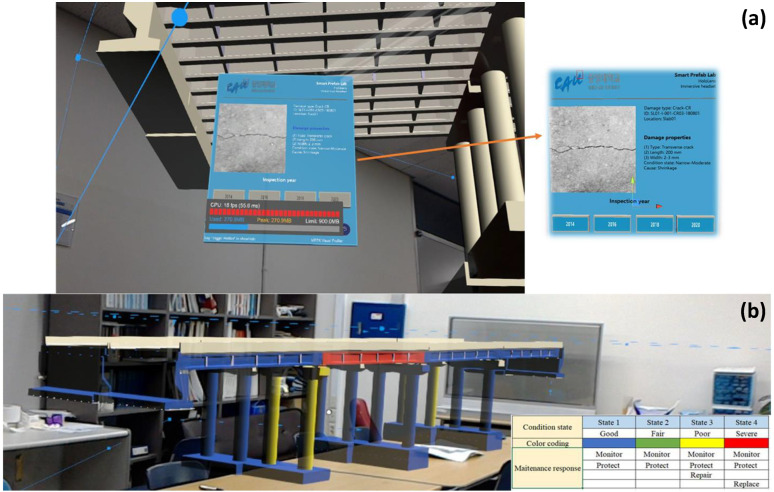
(**a**) Damage visualization and (**b**) condition rating visualization of bridge structure in HoloLens [130].

**Figure 23 sensors-22-09560-f023:**
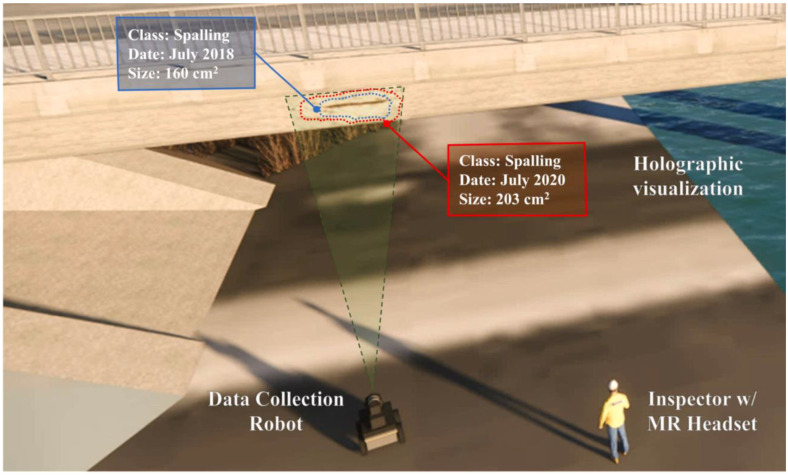
Descriptive visualization of HMCI for bridge inspection [131].

**Figure 24 sensors-22-09560-f024:**
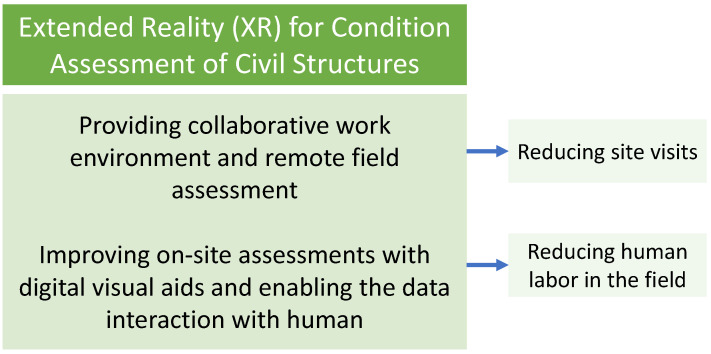
Highlights observed in XR studies presented in this paper.

**Figure 25 sensors-22-09560-f025:**
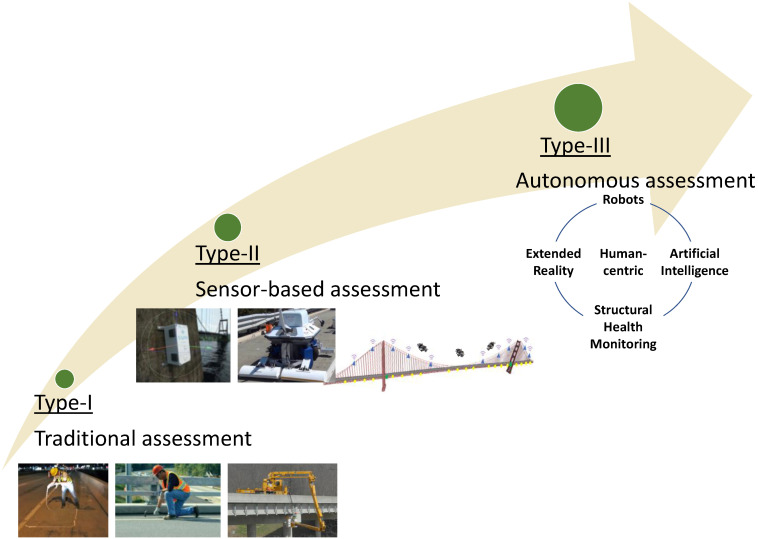
Roadmap for condition assessment of civil structures.

**Table 1 sensors-22-09560-t001:** General keywords used in the literature search for the Exploratory Review.

General Keywords X: Link the Words with “AND”	Civil Engineering	Civil Structures	AEC	Virtual Reality	Augmented Reality	Mixed Reality	Extended Reality
Civil Engineering				X	X	X	X
Civil Structures				X	X	X	X
AEC				X	X	X	X
Virtual Reality	X	X	X				
Augmented Reality	X	X	X				
Mixed Reality	X	X	X				
Extended Reality	X	X	X				

**Table 2 sensors-22-09560-t002:** On-target keywords used in the literature search for the Refined Review.

On-Target KeywordsX: Link the Words with “AND”	Condition Assessment	Inspection	Structural Health Monitoring	Non-Destructive Technique/Evaluation	Virtual Reality	Augmented Reality	Mixed Reality	Extended Reality
Condition Assessment					X	X	X	X
Inspection					X	X	X	X
Structural Health Monitoring					X	X	X	X
Non-Destructive Technique/Evaluation					X	X	X	X
Virtual Reality	X	X	X	X				
Augmented Reality	X	X	X	X				
Mixed Reality	X	X	X	X				
Extended Reality	X	X	X	X

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
