# Peer review of "Extended Reality (XR) for Condition Assessment of Civil Engineering Structures: A Literature Review"

_sensors, 2022, doi:10.3390/s22239560_

Round 1
Reviewer 1 Report
In this manuscript, the authors reviewed the research progress of AR/VR/MR (XR) explorations and applications in the architectural, engineering and construction (AEC) industry. The review fully explained the motivations to introducing XR in AEC, the definition of XR, the current progress published in different form factors and provided the recommendations for future development. The literature summarized the major contributions for different papers and provided a comprehensive view for readers to understand. Overall, the review is well-organized and in great quality. I have the following minor suggestions and comments that hope the authors could address.
1. Vuzix is one of the major players in the AR industry. And more specifically, they are developing HMDs for construction workers. It may be worth some discussions for the authors to talk about the status of Vuzix product and function to help readers understand the industry better.
2. If possible, besides journal papers, there are lots of industry development going on like Mecedes with HoloLens 2. Those also play the major role for generalization of XR to the industry. Hope the authors could cover some of them as a part of info besides journal review.
3. The formatting of the manuscript needs to be checked.
Author Response
The authors thank Reviewer#1 for encouraging and constructive remarks. The review is carefully considered, and the revised version of the manuscript addresses the points the reviewer raised. The revised sentences are yellow-highlighted both in this document and in the manuscript.
Reviewer#1: In this manuscript, the authors reviewed the research progress of AR/VR/MR (XR) explorations and applications in the architectural, engineering and construction (AEC) industry. The review fully explained the motivations to introducing XR in AEC, the definition of XR, the current progress published in different form factors and provided the recommendations for future development. The literature summarized the major contributions for different papers and provided a comprehensive view for readers to understand. Overall, the review is well-organized and in great quality. I have the following minor suggestions and comments that hope the authors could address.
Comment 1: Vuzix is one of the major players in the AR industry. And more specifically, they are developing HMDs for construction workers. It may be worth some discussions for the authors to talk about the status of Vuzix product and function to help readers understand the industry better.
Authors:
The authors appreciate the Reviewer’s suggestion. The authors mentioned the “smart glasses from Vuzix” in Section 2, along with the other products. The scope of this literature review study is defined as presenting the studies conducted using XR tools for condition assessment of civil engineering structures. In this regard, some of the XR tools used in those studies and some HMDs are mentioned in Section 2. Further discussion and comparative analysis of XR products in the market are not within the scope of this paper. It is our intention to inform the readers about the studies conducted to date in the literature using XR.
Comment 2: If possible, besides journal papers, there are lots of industry development going on like Mecedes with HoloLens 2. Those also play the major role for generalization of XR to the industry. Hope the authors could cover some of them as a part of info besides journal review.
Authors:
The authors thank the Reviewer for the suggestion. As mentioned, the scope of this literature review study is defined as presenting the studies conducted using XR tools for condition assessment of civil engineering structures. The authors intend to inform the readers about the completed works using XR technologies for condition assessment of civil engineering structures. Nevertheless, in the conclusion section of this paper, a comparative analysis study on using different XR tools for condition assessment of civil structures is suggested as a future work, which is also quoted below.
“Future studies can be explored on comparative analysis of VR/AR/MR tools, such as different HMDs, for using them for the condition assessment of civil structures. In this regard, each HMD can be listed in terms of its use efficiency for various purposes.”.
Comment 3. The formatting of the manuscript needs to be checked.
Authors:
The manuscript's formatting is checked, and additional proofreading is conducted. The authors hope that the manuscript is now suitable for publication
Reviewer 2 Report
F. N. Catbas et al. have presented a literature review yet encompassing the utilization of XR technologies for the condition assessment of civil structures. The presented investigation targets to provide essential information and guidelines for practitioners
and researchers about using XR technologies. The manuscript is well organized and informative. In my best opinion, this manuscript can be accepted after minor revision. The reviewer's comments are below:
- The titles of table 1 and 2 are not clear. Please, fix the writing orientation.
- I recommend to considering even more the role and impact of Artificial Intelligence (AI).
Author Response
The authors thank Reviewer#2 for detailed and insightful remarks. The review is carefully considered, and the revised version of the manuscript addresses the points the reviewer raised. The revised sentences are turquoise-highlighted both in this document and in the manuscript.
Reviewer#2: F. N. Catbas et al. have presented a literature review yet encompassing the utilization of XR technologies for the condition assessment of civil structures. The presented investigation targets to provide essential information and guidelines for practitioners and researchers about using XR technologies. The manuscript is well organized and informative. In my best opinion, this manuscript can be accepted after minor revision. The reviewer's comments are below:
Comment 1: The titles of table 1 and 2 are not clear. Please, fix the writing orientation.
Authors:
The authors would like to thank the Reviewer for the meticulous comment. The goal behind Table 1 and 2 is to assist the readers in finding relevant studies using such words. Table 1 displays more general keywords related to XR and Civil Engineering, which was the initial effort for the literature review. Table 2, on the other hand, shows more specific keywords related to the condition assessment of the civil structures. The titles of both tables are now modified in the manuscript and also shown below per the Reviewer’s suggestion.
“Table 1. General keywords used in the literature search for the Exploratory Review.”.
“Table 2. On-target keywords used in the literature search for the Refined Review.”.
Comment 2: I recommend to considering even more the role and impact of Artificial Intelligence (AI).
Authors:
The authors appreciate the constructive suggestion. The role of AI is indeed very critical. The authors defined the boundaries of this study within XR applications used for condition assessment of civil structures. Thus, the main focus of this study was to present the studies conducted using XR for condition assessment of civil structures. Nevertheless, the position of AI in XR is discussed in Section 4.3, where the recommendations and current and future trends of using XR tools for condition assessment of civil structures are mentioned, e.g., Fig. 25. Additionally, there were several studies presented in this literature paper that employ AI in their frameworks, such as the studies cited in this literature review work [10-11]. Additionally, the manuscript's formatting is checked, and additional proofreading is conducted. The authors hope that the manuscript is now suitable for publication.